# Population-based epidemiological analysis of acute pyelonephritis and antibiotic prescription in Spain (2009–2018)

Jesús Redondo-Sánchez[1,2☺], Ricardo Rodríguez-Barrientos[2,3,4☺] *,
Cristina Muntañola-Valero[2,3,5], Mª del Canto de-Hoyos-Alonso[2,6], Nuria Echave-Heras[7],
Lucia Martínez-Manrique[8,9], Miguel Gil-García[10], Isabel del Cura-González[2,3,4,11,12☺]

1 Ramon y Cajal Health Care Centre, Alcorcón. Primary Care Management. Servicio Madrileño de Salud, Madrid, Spain, 2 Network for Research on Chronicity, Primary Care, and Health Promotion (RICAPPS), Spain, 3 Research Unit, Primary Care Management, Madrid Health Service, Madrid, Spain, 4 Gregorio Marañón Health Research Institute IiSGM, Madrid, Spain, 5 Foundation for Biosanitary Research and Innovation in Primary Care (FIIBAP, Madrid, Spain, 6 Pedro Laín Entralgo Health Care Center, Alcorcón, Primary Care Management, Madrid Health Service, Madrid, Spain, 7 National School of Public Health, Carlos III Institute of Health, Madrid, Spain, 8 Preventive Medicine Department, Hospital Universitario de Móstoles, C. Dr. Luis Montes, S/N, Madrid, Spain, 9 Public Health and Epidemiology Research Group, School of Medicine, University of Alcalá, Madrid, Spain, 10 BIFAP (Database for Pharmacoepidemiological Research in the Public Domain), 11 Department of Medical Specialties and Public Health. Rey Juan Carlos University, Alcorcón, Madrid, Spain, 12 Aging Research Center, Karolinska Institutet and Stockholm University, Stockholm, Sweden

☺ "These authors contributed equally to this work."
* ricardo.rodriguez@salud.madrid.org

## Abstract

### Objectives

To estimate the incidence of pyelonephritis in primary care in Spain from 2009 to 2018, assess the associated antibiotic prescriptions, and analyse trends by sex and age.

### Method

This is a retrospective observational population-based national study using the Database for Pharmacoepidemiological Research in the Public Domain (BIFAP), which contains primary care electronic medical records, and is representative of the Spanish population. Patients with a diagnosis of pyelonephritis were included. Socio-demographic and clinical data were collected. Crude and adjusted incidence rates were calculated per 10,000 person-years by sex and age, and annual and global percentages of antibiotic use were calculated by sex, age, and antibiotic group. Trend analysis was performed using a joinpoint regression model.

### Results

24,888 cases of pyelonephritis were recorded with an incidence of 4.2/10,000 person-years (6.6 women vs 1.5 men). An annual decreasing trend was observed in women

**Data availability statement:** Availability of data and materials: Ethics Committee of the Hospital Universitario Fundación Alcorcón has approved this research, including any potential data sharing. For the study data that were used and analyzed, please contact the Research Unit of the Primary Care Management of Madrid: uinvestigacion.ap@salud.madrid.org All data generated or analyzed during this study are included in this published article [and its supplementary information files].

**Funding:** This study is funded by the Instituto de Salud Carlos III (ISCIII) through project PI19/01700 and co-funded by the European Union. The funders had no role in study design, data collection and analysis, decision to publish, or preparation of the manuscript.

**Competing interests:** The authors have declared that no competing interests exist.

**Abbreviations:** AEMPS, Spanish Agency for Medicines and Medical Devices; AAPC, Average annual percentage change; APC, Annual percentage change; BIFAP, Database for Pharmacoepidemiological Research in the Public Domain; CIs, confidence intervals; E. Coli, Escherichia coli; ICPC-2, International Classification of Primary Care; IQR, Interquartile range; NHS, National Health Service; RECORD, Reporting of Studies Conducted using Observational Routinely collected Health Data; UTIs: Urinary tract infections.

(AAPC average annual percentage change) −2.7 (95% CI −4.4;-0.9), men −3.0 (95% CI −4.5;-1.5), patients aged 18–64 years −2.9 (95% CI −4.7;-0.9) and ≥ 65 years −4.2 (95% CI −5.8;-2.4). The most frequently used groups of antibiotics were cephalosporins (38.7%), quinolones (30%), combined penicillins (22.2%) and fosfomycin (6.8%). Cephalosporin prescription predominated in women (39.8%), and quinolone prescription in men (40.3%). The most prescribed cephalosporins were third and second generation (21% and 17.7% respectively). A downward annual trend was observed in the global use of antibiotics AAPC −0.8 (95% CI −1.4; −0.2), with an increasing AAPC 1.7 (95% CI −0.6; 3.4) in ≥65 years. Among the groups of antibiotics, the prescription of quinolones AAPC −6.9 (95% CI −31.7;17.3) and penicillins AAPC −8.7 (95% CI −11.2;-6.8) decreased and cephalosporins increased AAPC 19.0 (95% CI 12.2;26).

## Conclusions

Women had a fourfold higher incidence of acute pyelonephritis than men, with a decreasing trend over the study period. Cephalosporins were the most commonly prescribed antibiotics in women, while quinolones were more common in men. An increasing trend in cephalosporin use and a decreasing in quinolone use were observed.

## Introduction

Urinary tract infections (UTIs) are among the most common bacterial infections in community and hospital settings [1]. These infections are an important reason for consultation in primary care and for the use of antimicrobials and are the second-leading infectious cause of hospitalization after respiratory infections [2].

Acute pyelonephritis is a UTI of the kidneys that accounts for 5.5% of all UTIs treated in primary care [3] and 14% to 52% of all admissions for UTIs [4–6].

Most patients with pyelonephritis are treated in primary care or in an outpatient setting, but patients with the most severe cases (14% to 25%) require hospitalization [7,8]. Consultations [3] and hospitalizations [9] are more common in infants, young women, and elderly individuals. For cases of pyelonephritis managed on an outpatient basis, the incidence is 4–10 times greater in women, with highly variable incidence rates per 10,000 inhabitants that range from 12 to 54 cases in women compared with 2–11 cases in men [7,8,10]. Hospitalization for pyelonephritis is also more common in women, with incidences per 10,000 inhabitants between 3 and 12 cases in women and 1 and 7 cases in men [8,11,12].

Pyelonephritis generally responds well to antibiotic treatment. However, pyelonephritis can sometimes lead to severe urinary sepsis, chronic kidney disease, and even death [13]. Advanced age, the presence of metastatic cancer or septic shock, treatment with corticosteroids and admission to intensive care units are risk factors for mortality in patients with UTIs [14]. In several studies of pyelonephritis, higher mortality rates have also been reported in men [10–12].

With respect to the etiology of pyelonephritis, *Escherichia coli* is detected in most cases that occur in young and healthy women, whereas in older women, men and institutionalized or immunosuppressed patients, other gram-negative bacteria, such as *Klebsiella pneumoniae*, *Proteus mirabilis* and *Pseudomonas aeruginosa*, are detected. There is a significant difference in causative bacteria depending on age, comorbidities and the location of infection [13,15].

The use of antimicrobials is one of the main factors responsible for antibiotic resistance [16]. Antibiotic resistance in *E. coli* is increasing in Europe [17]. Spain, along with other southern European countries, has one of the highest rates of resistance to antibiotics. In 2020, the rate of *E. coli* resistance to amoxicillin/ampicillin was 57%, whereas the rates of resistance to fluoroquinolones and third-generation cephalosporins were 28.6% and 14.1%, respectively [17]. Knowledge of models of antimicrobial prescription and their temporal changes is essential to correctly evaluate antimicrobial prescription and propose actions for improvement. This information can help both health-care physicians and health-care authorities [18] adapt guidelines for antimicrobial use and improve the empirical treatment of this group of infections. This work is part of a line of research conducted by our group on UTIs in the primary care and hospital settings [4,5,19].

The objectives of this study were to estimate the incidence of acute pyelonephritis in primary care in Spain between 2009 and 2018, determine the associated prescription of antibiotics, and analyze these trends according to sex and age.

## Materials and methods

A retrospective observational study was conducted using Database for Pharmacoepidemiological Research in the Public Domain (BIFAP), which is the database of the National Health System of Spain and is managed by the Spanish Agency for Medicines and Health Products (AEMPS) in collaboration with the autonomous communities that participate in the project. BIFAP is a population-based database that includes anonymized data from the electronic medical records of all patients seen in family medicine and pediatric primary care clinics since 2001. The database is updated periodically, at least once a year, with data provided by each of the participating autonomous communities. BIFAP has been used in other studies of disease incidence and drug trends. Currently, BIFAP includes data from 15,373 general practitioners and paediatricians quotas in 12 autonomous communities—out of a total of 17 in Spain—covering 22.5 million patients, whose distribution by age and sex is comparable to that of the Spanish population. Participation by autonomous communities is voluntary. The mean follow-up of the patients included in BIFAP is 10.67 years, totaling 240 million person-years of follow-up. The database includes 1 billion health problem records and 3.3 billion medication records (http://www.bifap.org; accessed January 27, 2025). As of 2018, the last year of the study period, BIFAP included anonymized data from 12 million patients, representing 17.3% of the Spanish population, with an average follow-up of 8.6 years (102 million person-years) [20].

The data were accessed for research purposes the 30 April 2021. We didn´t have access to information that could identify individual participants during or after data collection.

The Reporting of Studies Conducted using Observational Routinely Collected Health Data (RECORD checklist) is available as supporting information; see S1 File.

This study included data from 8,615,690 people over 18 years of age registered in BIFAP from January 1, 2009, to December 31, 2018 (59,802,261 million person-years of follow-up with a mean follow-up of 6.9 years). Patients who had episodes of acute pyelonephritis (recorded with the International Classification of Primary Care (ICPC-2) code U70) for which an antibiotic was prescribed or for whom a urine culture was requested on the day of diagnosis were identified. The crude incidence rates of pyelonephritis per 10,000 population-years by for sex and age groups (18–64 years and 65 years or older) along with the corresponding 95% confidence intervals (CIs) were calculated.

Regarding the prescription of antibiotics, first, the percentage of prescribed antibiotics (all antibiotic groups and the most commonly prescribed active principles) (suplemento 2) among the total number of antibiotics prescribed during the study period was determined by age group and sex. Second, to analyze the evolution of prescriptions, the most prescribed antibiotic groups for pyelonephritis (specifically, the number of prescriptions per 100 cases of pyelonephritis) were determined according to sex and age groups.

Analysis of the temporal evolution of the incidence of acute pyelonephritis and the use of antibiotics during the study period was conducted via joinpoint regression. Joinpoint regression models are used to identify the points where there are significant changes in the trend of rates throughout a period. This approach has been widely used by fitting a model when the junction points (i.e., the points of change, inflection points, or joinpoints) are not known a priori and must be estimated from the data.

The Joinpoint software identifies change points without user input by fitting segmented (piecewise) regression models to time series data, where each segment is defined by a distinct slope, and the joinpoints represent statistically significant changes in trend. The process does not require the user to predefine the number or location of joinpoints. Instead, it uses model selection procedures such as the Bayesian Information Criterion (BIC) to determine the optimal number of joinpoints, evaluating the statistical significance of each change in slope. In our analysis, we used the default settings, including the option to restrict joinpoints to occur at observed time points rather than between them. This allowed us to identify the specific years in which trend changes occurred. The software provides both graphical representations and detailed listings of the segments (years, in our study) for each analysis, based on the user-defined parameters and statistical adjustments.

For the pyelonephritis trend analysis, the rates of pyelonephritis per 100,000 inhabitants were adjusted to the 2014 BIFAP population. Antibiotic prescription trend analysis was performed per 100,000 cases of pyelonephritis. The annual percentage change (APC) was estimated in each linear segment, and the average annual percentage change (AAPC) was calculated as a weighted average of the APC of the model along with the corresponding 95% CIs [21].

### Ethical aspects

The protocol was approved by the Scientific Committee of BIFAP on January 24, 2019. The study was approved by the Hospital Fundación Alcorcon Ethics Research Committee (dated March 25, 2021) of the Central Research Commission of the Primary Health Care Management of Madrid (Code 13/19). The requirement to obtain consent was waived by the ethics committee (dated March 25, 2021). The data were fully anonymized by BIFAP before they were accessed by investigators. This study adheres to the fundamental ethical principles of autonomy, beneficence, justice, and non-maleficence. It follows the standards of Good Clinical Practice and the guidelines outlined in the most recent Declaration of Helsinki (2024).

## Results

### Incidence of acute pyelonephritis

Between 2009 and 2018, a total of 24,888 cases of acute pyelonephritis were registered. Of the total cases, 72.5% occurred in women aged 18–64 years (Table 1).

The overall incidence of pyelonephritis in the study period was 4.2 cases per 10,000 person-years of follow-up. The incidence was greater in women than in men (6.6 vs 1.5). In women, the highest incidence was in the group aged 18–64 years (7.8 cases vs. 3.3 in those over 65 years), whereas in men, the incidence was higher in those aged ≥ 65 years (2.0 in those aged ≥65 years; 1.3 in those aged 18–64 years). This age and sex distribution was maintained throughout the study period. From 2009 to 2018, the global incidence of pyelonephritis decreased from 4.9 to 3.6 cases per 10,000 population-years (Table 1). The decrease occurred in both men (from 1.8 to 1.2) and women (from 7.7 to 5.9) and in both elderly individuals and those under 65 years of age (Fig 1).

Analysis of the trend in the incidence of pyelonephritis with the joinpoint regression model (Fig 1) showed that the downward trend was significant both globally, with an AAPC value of −3.5 (95% CI −5.3; −1.6), and in both sexes and age groups, with unique trends throughout the period. The trend analysis of the global incidence, adjusted for sex and age and specific to age and sex groups, is presented in Table 2.

**Table 1. Rates of acute pyelonephritis per 10,000 inhabitants according to sex and age group in the population > 18 years of age in Spain, 2009-2018.**

| Year | TOTAL | | MEN | | | | | | WOMEN | | | | | |
|---|---|---|---|---|---|---|---|---|---|---|---|---|---|---|
| | | | Total men | | 18-64 years | | ≥65 years | | Total women | | 18-64 years | | ≥65 years | |
| | n | I | n | I | n | I | n | I | n | I | n | I | n | I |
| 2009 | 2205 | 4.9 | 367 | 1.8 | 267 | 1.6 | 100 | 2.7 | 1838 | 7.7 | 1605 | 8.8 | 233 | 4.1 |
| 2010 | 2456 | 5.2 | 390 | 1.8 | 288 | 1.6 | 102 | 2.5 | 2066 | 8.1 | 1829 | 9.5 | 237 | 3.9 |
| 2011 | 2434 | 5.1 | 417 | 1.9 | 307 | 1.7 | 110 | 2.6 | 2017 | 7.9 | 1765 | 9.1 | 252 | 4.0 |
| 2012 | 2427 | 3.9 | 425 | 1.5 | 303 | 1.3 | 122 | 2.0 | 2002 | 6.0 | 1726 | 7.0 | 276 | 3.1 |
| 2013 | 2564 | 3.9 | 446 | 1.5 | 308 | 1.3 | 138 | 2.2 | 2118 | 6.1 | 1857 | 7.3 | 261 | 2.8 |
| 2014 | 2679 | 4.1 | 447 | 1.4 | 312 | 1.3 | 135 | 2.1 | 2232 | 6.4 | 1926 | 7.6 | 306 | 3.2 |
| 2015 | 2668 | 4.0 | 449 | 1.4 | 311 | 1.3 | 138 | 2.0 | 2219 | 6.3 | 1889 | 7.4 | 330 | 3.4 |
| 2016 | 2637 | 4.0 | 407 | 1.3 | 296 | 1.2 | 111 | 1.6 | 2230 | 6.4 | 1934 | 7.7 | 296 | 3.0 |
| 2017 | 2505 | 3.8 | 388 | 1.3 | 257 | 1.1 | 131 | 1.9 | 2117 | 6.2 | 1844 | 7.5 | 273 | 2.8 |
| 2018 | 2313 | 3.6 | 358 | 1.2 | 255 | 1.1 | 103 | 1.5 | 1955 | 5.9 | 1680 | 7.1 | 275 | 2.9 |
| 2009-2018 | 24888 | 4.2 | 4094 | 1.5 | 2904 | 1.3 | 1190 | 2.0 | 20794 | 6.6 | 18055 | 7.8 | 2739 | 3.3 |

n: number of cases, I: incidence rate

## Antibiotics prescribed for acute pyelonephritis

During the study period, the most widely used group of antibiotics was cephalosporins (38.7%), followed by quinolones (30%), penicillin derivatives (21.4%) and fosfomycin (6.8%). The remaining antibiotics were used in fewer than 1.1% of total cases. Supplement 2 provides a more detailed breakdown of the usage percentages of each active principle analyzed in relation to the total number of antibiotics prescribed.

In women, the prescription of cephalosporins (39.8%) predominated over the prescription of quinolones (28%), whereas in men, the prescription of quinolones (40.3%) surpassed the prescription of cephalosporins (32.8%). In both sexes, the use of penicillin derivatives was greater in people aged 18–64 years, whereas fosfomycin was used more often in people aged ≥65 years. Trimethoprim sulfamethoxazole (TMP_SMX) was used more often in men (1.3%) than in women (0.7%), and in both sexes, it was used more often in older people (Table 3).

The most frequently prescribed antibiotics in each group with respect to the total number of antibiotics prescribed are shown by sex and age groups in Fig 2.The most prescribed cephalosporins were third- and second-generation (21% and 17.7%, respectively). The most prescribed antibiotic in the penicillin group was amoxicillin-clavulanate (21.4%). The most prescribed quinolone was ciprofloxacin (22.7%), followed by levofloxacin (4.5%) and norfloxacin (2.3% of all antibiotics). Supplement 2 provides a more detailed breakdown of the usage percentages of each active ingredient analyzed in relation to the total prescribed antibiotics.

With regard to changes in antibiotic prescription from 2009 to 2018 for every 100 cases of acute pyelonephritis, the use of cephalosporins increased (from 33.1 to 57.3% in women and from 23.7 to 46.6% in men), the use of quinolones decreased (from 54.2 to 33.5% in men and from 44.8 to 25.4% in women), and the use of penicillin derivatives decreased (from 34.6 to 16% in women and from 30.5 to 13.4% in men). Throughout the entire period, quinolones were used more often in men, whereas cephalosporins, penicillin derivatives and fosfomycin were used more often in women (Fig 3). Compared with the other age groups, quinolones and fosfomycin were prescribed more frequently for people older than 64 years, whereas in the 18- to 64-year age group, the most frequently prescribed antibiotics were cephalosporins and penicillin combinations (Fig 4).

When the trends were checked with joinpoint regression, the overall number of prescriptions of antibiotics decreased significantly during the study period [AAPC −0.8; (95% CI −1.4; −0.2)] in both women [AAPC −0.8 (95% CI −1.3; −0.2)]

**(A)**

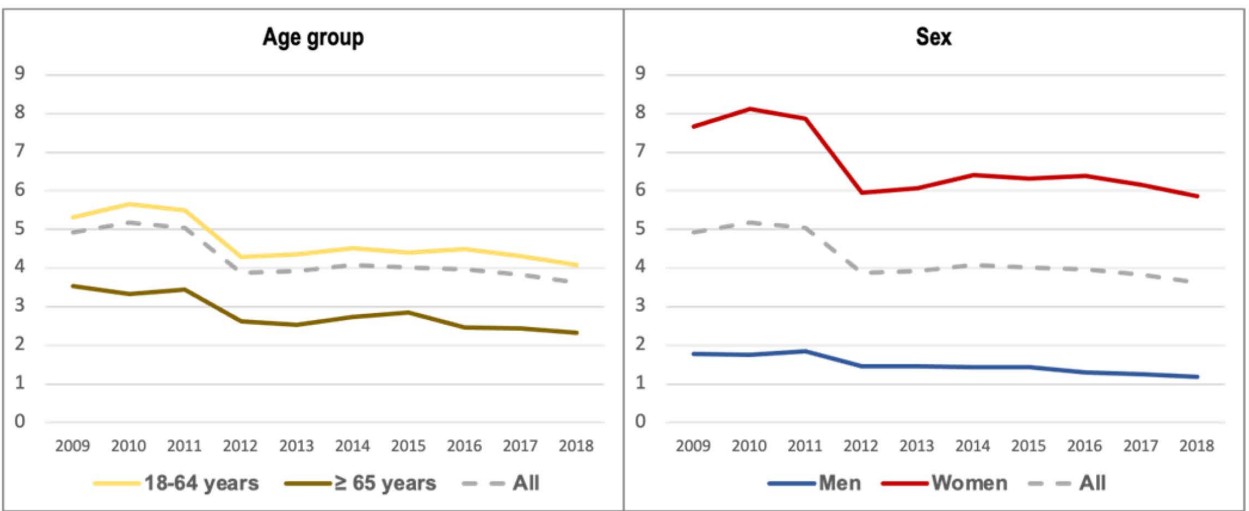

**(B)**

| | Age groups | | Sex | | All |
|---|---|---|---|---|---|
| | 18 to 64 years | ≥ 65 years | Women | Men | |
| **AAPC (2009-2018)** | -3.1** (95%CI -5,0; -1.2) | -4.7** (95%CI -6.7; -2.6) | -3.1** (95%CI -4.8; -1.3) | -4.7** (95%CI -5.9; -3.5) | -3.5** (95%CI -5.3; -1.6) |
| **APC** | (2009-2018) -3.1** (95%CI -5.0; -1.2) | (2009-2018) -4.7** (95%CI -6.7; -2.6) | (2009-2018) -3.1** (95%CI -4.8; -1.3) | (2009-2018) -4.7** (95%CI -5.9; -3.5) | (2009-2018) -3.5** (95%CI -5.3; -1.6) |

**Fig 1. Incidence trends of acute pyelonephritis by age group and sex in the population > 18 years of age in Spain, 2009-2018. (A)** Representation of the incidence of pyelonephritis per 10,000 population throughout the study period. **(B)** Joinpoint analysis of the global trend and according to sex and age group. AAPC: average annual percentage change, APC: annual percentage change, CI: confidence interval, **: statistically significant.

and men [AAPC −1.2 (95% CI −2.0; −0.4)]. This decrease was significant in the group aged 18−64 years [AAPC −0.9 (95% CI −1.3; −0.4)] and was more pronounced between 2016 and 2018 [APC −2.5 (95% CI −4.6; −0.4)]. In people older than 64 years, there was an increase in prescriptions that was significant only from 2011 to 2018 [APC 14.7 (95% CI 2.9; 25.0)] (Table 4).

Regarding the trend of the most frequently prescribed antibiotic groups, the use of cephalosporin increased [AAPC 19.0 * (95% CI 12.2; 26.0)], while quinolone [AAPC −6.9 (95% CI −31.7; 17.3)] and penicillin [AAPC −8.7 (95% CI −11.2; −6.8)] prescriptions decreased significantly. Fosfomycin prescriptions increased nonsignificantly [AAPC 2.5 (95% CI −0.8; 6.0)] (Table 4). These global trends were maintained in both sexes and different age groups, with the exception of fosfomycin, for which prescriptions decreased in men (Figs 3 and 4).

## Discussion

The incidence of acute pyelonephritis of community origin and the use of antibiotics decreased from 2009 to 2018. The most widely used antibiotics were cephalosporins, quinolones and amoxicillin-clavulanate, with different profiles according to sex and age. During the study period, the use of quinolones decreased, whereas the use of cephalosporins increased.

**Table 2. Analysis of trends in the incidence of acute pyelonephritis per 100,000 inhabitants in the population > 18 years of age in Spain, 2009–2018.**

| | | AAPC (2009-2018) | APC |
|---|---|---|---|
| Global | | -3.5* (95% CI -5.3; -1.6) | (2009-2018) -3.5* (95% CI -5.3; -1.6) |
| Adjusted for sex | 18 to 64 years | -2.9* (95% CI -4.7; -0.9) | (2009-2018) -2.9* (95% CI -4.7; -0.9) |
| | ≥ 65 years | -4.2* (95% CI -5.8; -2.4) | (2009-2018) -4.2* (95% CI -5.8; -2.4) |
| Adjusted for age | Women | –2.7* (95% CI -4,4; -0,9) | (2009-2018)–2.7* (95% CI -4.4; -0.9) |
| | Men | -3.0* (95% CI -4.5; -1.5) | (2009-2015) -7.9* (IC 95% -11; -5.8) (2015-2018) 7.5 (IC 95% 1.5; -16.9) |
| Specific | Women 18-64 years | -2.6* (95% CI -4.6; -0.4) | (2009-2018) -2.6* (95% CI -4.6; -0.4) |
| | Women ≥ 65 years | -3.6* (95% CI -5.7; -1.4) | (2009-2018) -3.6* (95% CI -5.7; -1.4) |
| | Men 18-64 years | -2.6* (95% CI -4.6; -0.7) | (2009-2015) -7.8* (95% CI -12.6; -5.2) (2015 a 2018) 8.7* (95% CI 0.9; 21.3) |
| | Men ≥ 65 years | -4.1* (95% CI -7.8; -0.6) | (2009-2015) -8.0 (CI95% -22.3; 10.0) (2015-2018) 4.4 (95% CI -10.7; 24.6) |

AAPC: average annual percentage change, APC: annual percentage change, CI: confidence interval.

* statistically significant

**Table 3. Prescription of antibiotic groups for acute pyelonephritis according to sex and age in the population > 18 years of age in Spain, 2009–2018.**

| | | Men | | | Women | | |
|---|---|---|---|---|---|---|---|
| Antibiotics group | Total (%) | Total (%) | 18-64 years (%) | ≥ 65 years (%) | Total (%) | 18-64 years (%) | ≥ 65 years (%) |
| Cephalosporins | 38.7 | 32.8 | 32.7 | 33.2 | 39.8 | 40.1 | 37.8 |
| Quinolones | 30.0 | 40.3 | 40.7 | 39.5 | 28.1 | 27.4 | 32.8 |
| Penicillin_combined | 22,2 | 19,5 | 20,2 | 17,9 | 22,7 | 23,5 | 17,2 |
| Fosfomycin | 6.8 | 4.5 | 4.0 | 5.7 | 7.2 | 6.9 | 9.6 |
| TMP/SMX | 0.8 | 1.3 | 1.2 | 1.8 | 0.7 | 0.7 | 1.1 |
| Aminoglycosides | 0,7 | 0.7 | 0.8 | 0.3 | 0.7 | 0.7 | 0.5 |
| Macrolides | 0.4 | 0.6 | 0.3 | 1.2 | 0.4 | 0.3 | 0.5 |
| Nitrofurantoin | 0.4 | 0.3 | 0.2 | 0.5 | 0.4 | 0.4 | 0.6 |

TMP/SMX: trimethoprim/sulfamethoxazole

Antibiotic prescriptions were given on Day 0 (at the time of pyelonephritis diagnosis).

(%) Percentages of each group in terms of the total number of antibiotics prescribed over 10 years

During the ten years of the study, we calculated a global incidence rate of pyelonephritis of 4.2 cases per 10,000 inhabitants. This rate was lower than that reported in other countries, such as the US and Korea [7,8,10], also calculated from databases.

Acute pyelonephritis was four times more common in women than in men, similar to the findings of other studies that revealed even greater differences [7,8]. With respect to age, in women, the incidence was greater in the 18- to 64-year-old group, whereas in men, it was greater in those older than 64 years. Acute pyelonephritis in young women has been associated with sexual activity (frequency of sexual intercourse in preceding days, use of spermicide or a new sexual partner) and a family and/or personal history of UTIs, and urinary incontinence and diabetes are factors related to hospital admission [22]. In men over 64 years of age, prostate disease and instruments used in urological procedures can contribute to an increase in pyelonephritis.

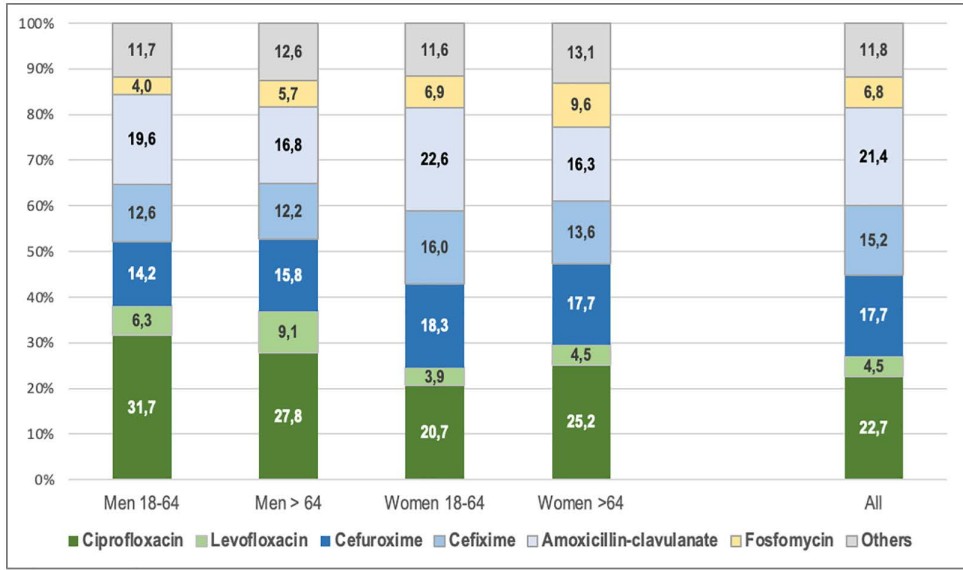

**Fig 2. Distribution by age and sex of the most prescribed antibiotics for acute pyelonephritis in the population > 18 years of age in Spain, 2009–2018.** Antibiotic prescriptions were made on Day 0 (at the time of diagnosis of pyelonephritis). Percentages of each antibiotic among the total number of antibiotics prescribed in the 10 years.

Throughout the study period, the overall incidence of acute pyelonephritis decreased slightly across both sexes and age groups, unlike other studies where the incidence rates of community pyelonephritis remained stable [8] or increased [7] as the number of consultations for acute pyelonephritis increased [3].This fact is difficult to interpret, although it may be related in part to the increase in hospitalization. In two previous publications on patients hospitalized for UTIs in Spain [4,5], we reported that the number of hospital admissions for pyelonephritis increased during the years of this study, especially in women aged 18–55 years [4]. This is partially explained by the occurrence of pyelonephritis in pregnant women; however, the trend was more stable in men and women aged 55–64 years or older than 64 years [4,5]. An increasing trend of admissions for pyelonephritis has also occurred in other countries, such as Korea [7], Denmark [9] and France [23]. In this last study, the authors reported an increase in urinary diversion procedures, such as double-J stents, which were associated with the presence of urolithiasis, comorbidities, increased age, sepsis, or UTIs caused by bacteria other than *Escherichia coli*. This may justify the increasing trend of hospitalizations for pyelonephritis.

There was a decrease in prescribed antibiotics in both sexes and age groups, unlike the findings of a Korean study [24] that found that antibiotic use in outpatients with pyelonephritis was stable or a Norwegian study [3] that found an increase in antimicrobial prescriptions. Notably, in our study, only antibiotics prescribed at the time of diagnosis were considered, while antibiotics prescribed later due to a lack of response or the appearance of resistance in the antibiogram were not considered. This may have influenced the difference in trends between this study and other works.

In our study, the antibiotics used most frequently for the treatment of acute pyelonephritis were cephalosporins followed by quinolones and penicillin derivatives. These three groups accounted for 90% of all prescriptions for pyelonephritis. In an older study from 1997–2001 in the US [8] and a more recent study in Korea (2010–2014) [24], quinolones were the first prescription, followed by cephalosporins. In a 10-year follow-up study conducted in Norway from 2006 to 2015 [3], the most commonly prescribed antibiotic was pivmecillinam, followed by ciprofloxacin and TMP-SMX. In Denmark [25], pivmecillinam was also the most prescribed antibiotic for upper UTIs, and the use of quinolones increased with patient age. Unlike northern Europe, where pivmecillinam is the first-line

**(A)**

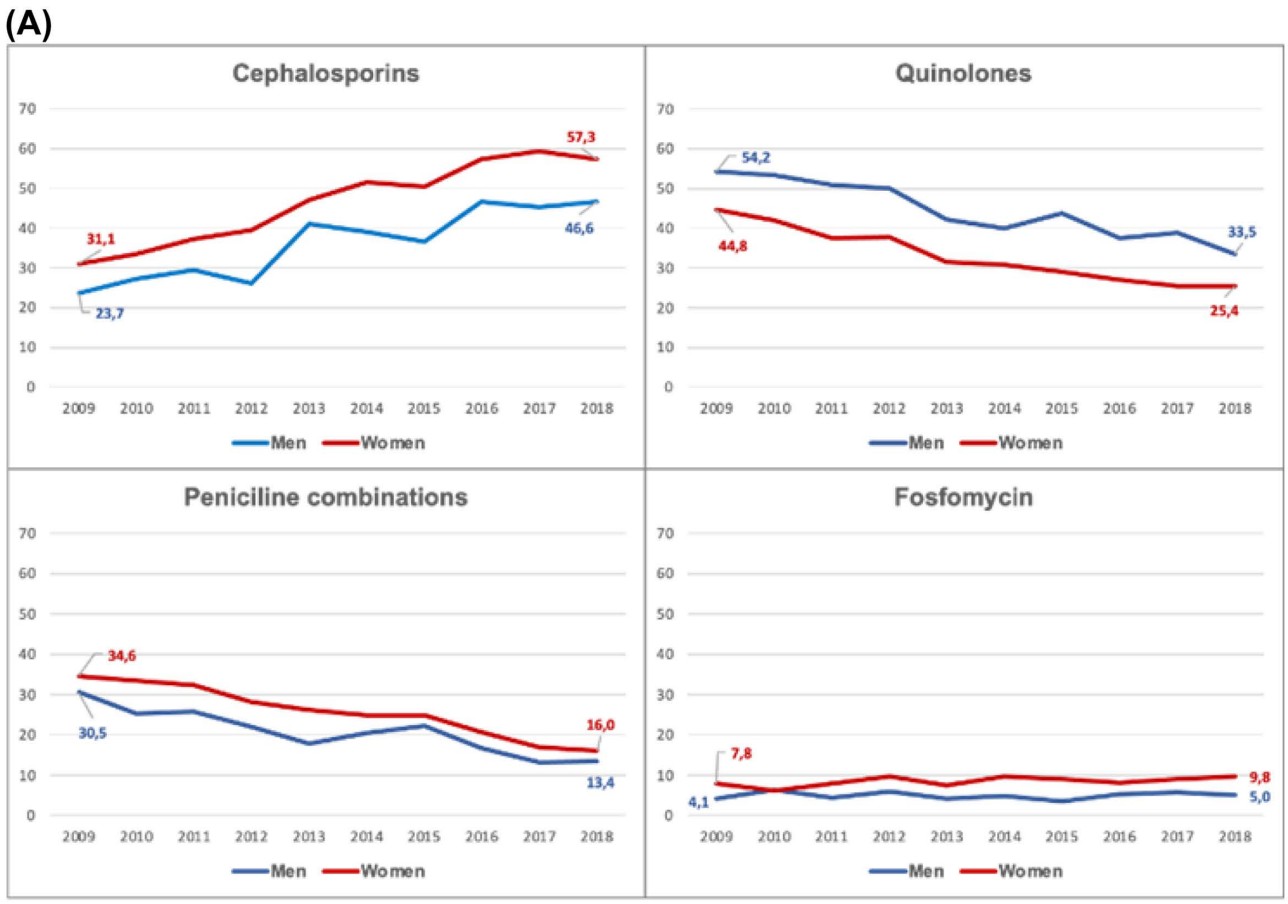

**(B)**

| | Cephalosporins | | Quinolones | | Penicillin combinations | | Fosfomycin | |
|---|---|---|---|---|---|---|---|---|
| | Men | Women | Men | Women | Men | Women | Men | Women |
| AAPC (2009-2018) | 7.9** (95% CI 4,4; 11.9) | 7.8** (95% CI 6.1; 10.1) | -5.0** (95% CI -7.1; -3.0) | -6.6** (95% CI -7.8; -5.5) | -7.8** (95% CI -11.0; -4.9) | -8.8** (95% CI -11.0; -7.1) | -0.2 (95% CI -6.9; 6.9) | 2.7 (95% CI -0.9; 6.6) |
| APC | 2009-2018 | 2009-2014 | 2009-2018 | 2009-2018 | 2009-2018 | 2009-2015 | 2009-2018 | 2009-2018 |
| | 7.8** (95% CI 4,4; 11.9) | 11.0** (95% CI 8.6; 21.7) | -5.0** (95% CI -7.1; -3.0) | -6.6** (95% CI -7.8; -5.5) | -7.8** (95% CI -11.0; -4.9) | -6.4 (95% CI -8.0; 0.0) | -0.2 (95% CI -6.9; 6.9) | 2.7 (95% CI -0.9; 6.6) |
| | | 2014-2018 3.9 (95% CI -4.2; 6.7) | | | | 2015-2018 -13.4** (95% CI -24.2; -9.1) | | |

**Fig 3. Most frequently prescribed antibiotic groups trends for acute pyelonephritis by sex in the population > 18 years of age in Spain, 2009–2018. (A)** Representation of the number of prescriptions per 100 cases of acute pyelonephritis throughout the study period. **(B)** Joinpoint analysis of the antibiotic prescriptions. AAPC: average annual percentage change, APC: annual percentage change, CI: confidence interval, **: statistically significant.

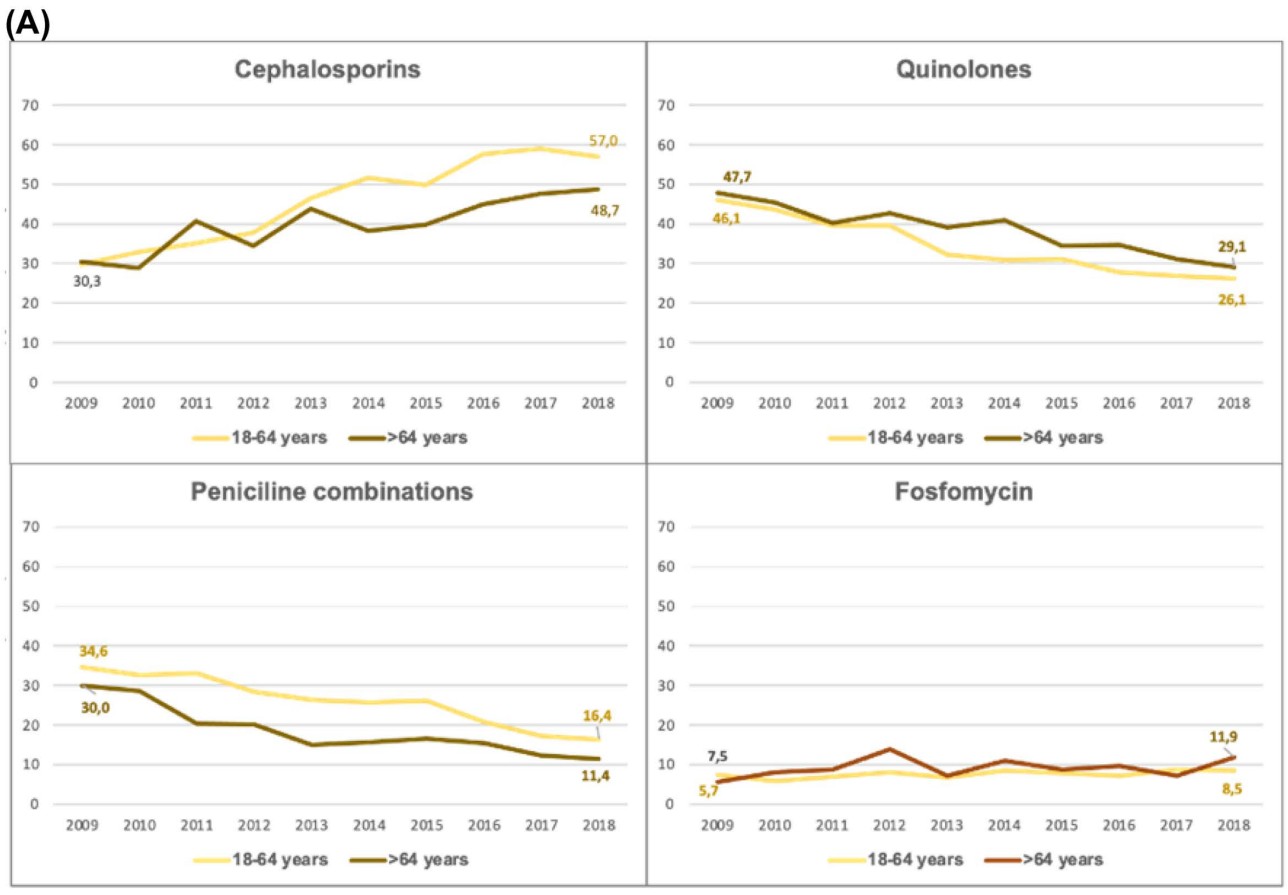

**(A)**

**(B)**

| | Cephalosporines | | Quinolones | | Penicillin combinations | | Fosfomycin | |
|---|---|---|---|---|---|---|---|---|
| | 18-64 años | >64 años | 18-64 años | >64 años | 18-64 años | >64 años | 18-64 años | >64 años |
| **AAPC** (2009-2018) | 8.4** (95% CI 6.8; 10.6) | 9.4** (95% CI 6.5; 13.0) | -6.6** (95% CI -7.7; -5.6) | -1.8 (95% CI -5.1; 0.6) | -8.6** (95% CI -11.0; -6.7) | -7. 5** (95% CI -12.3; -3.3) | 2.5 (95% CI -1.7; 7.1) | 7.2 (95% CI -4.8; 23.0) |
| **APC** | 2009-2014 11.9** (95% CI 9.4; 20.9) | 2009-2011 32.6** (95% CI 12.5; 55.2) | 2009-2018 -6.6** (95% CI -7.7; -5.6) | 2009-2011 13.0 (95% CI -1.00; 27.6) | 2009-2015 -5.6 (95% CI -7.5; 0.6) | 2009-2018 -7. 5** (95% CI -12.3; -3.3) | 2009-2018 2.5 (95% CI -1.7; 7.1) | 2009-2012 32.3 (95% CI -5.3; 179.3) |
| | 2014-2018 4.0 (95% CI -3.1; 7.1) | 2011-2018 3.5 (95% CI -1.6; 6.1) | | 2011-2018 -5.6** (95% CI -15.1; -3.3) | 2015-2018 -14.2** (95% CI -25.8; -9.1) | | | 2012-2018 -3.5 ((95% CI -40.7; 18.3) |

**Fig 4. Most-prescribed antibiotic groups trends for acute pyelonephritis by age group in the population > 18 years of age in Spain, 2009–2018.** **(A)** Representation of the number of prescriptions per 100 cases of acute pyelonephritis throughout the study period. **(B)** Joinpoint analysis of the antibiotic prescriptions. AAPC: average annual percentage change, APC: annual percentage change, CI: confidence interval, **: statistically significant.

antibiotic for the treatment of lower and upper acute UTIs [3,25], in Spain, The use of ~~cefuroxime or~~ third-generation cephalosporins is recommended as the first choice in patients with uncomplicated acute pyelonephritis [26–28]. At the time of the study, cefuroxime was also recommended [29,30]. Cephalosporins should not be used

**Table 4. Trend analysis of antibiotic prescriptions per 100,000 cases of pyelonephritis in the population > 18 years of age in Spain, 2009–2018.**

| | | AAPC (2009-2018) | APC | |
| --- | --- | --- | --- | --- |
| | | | Period | |
| Global | | -0,8* (95% CI -1,4; -0,2) | 2009-2018 | -0,8* (95% CI -1,4; -0,2) |
| Specific by age | Women | - 0,8* (95% CI -1,3; -0,2) | 2009-2018 | -0,8* (95% CI -1,3; -0,2) |
| | Men | -1,2* (95% CI-2,0; -0,4) | 2009-2018 | -1,2* (95% CI-2,0; -0,4) |
| Specific by sex | 18-64 years | -0.9* (95% CI -1.3; -0.4) | 2009-2016 | -0.4 (95% CI -1,2; 1.4) |
| | | | 2016-2018 | -2.5* (95% CI -4.6; -0.4) |
| | >65 years | 1.7 (95% CI -0.6; 3.4) | 2009-2011 | 14.7* (95% CI 2.9; 25.0) |
| | | | 2011-2018 | -1.8 (95% CI -7.3; 0.7) |
| By groups of antibiotics | Cephalosporins | 19.0* (95% CI 12.2; 26.0) | 2009-2016 | 6.8 (95% CI -15.81; 16.33) |
| | | | 2016-2018 | 73.9* (95% CI 29.5; 115.6) |
| | Quinolones | -6.9 (95% CI -31.7; 17.3) | 2009-2014 | 29.6* (95% CI 6.7; 192.2) |
| | | | 2014-2018 | -38.5* (95% CI -89.2; -17.7) |
| | Combined penicillin | -8.7* (95% CI -11.2; -6.8) | 2009-2015 | -6.4 (95% CI -8.5; 1.9) |
| | | | 2015-2018 | -13.3* (95% CI -25.1; -8.3) |
| | Fosfomycin | 2.5 (95% CI -0.8; 6.0) | 2009-2018 | 2.5 (95% CI -0.8; 6.0) |

AAPC: average annual percentage change, APC: annual percentage change; CI: confidence interval.

* statistically significant

empirically if the patient had a recent urine culture showing an ESBL-producing microorganism or if local resistance rates are high [28].

Our study found that the prescription rate of cephalosporins doubled from 2009 to 2018. Cephalosporins were the most prescribed antibiotics in women of all ages throughout the study period, predominantly in women aged 18–64 years. The most prescribed antibiotics in the cephalosporin group were cefuroxime (2nd generation) and cefixime (3rd generation). In Spain, the prevalence of resistance to this group is approximately 15% [17]; therefore, cephalosporins are a first-line treatment alternative [29].

Fluoroquinolones were used in one-third of cases, although their use decreased throughout the study period. In men, fluoroquinolones were the most widely used group of antibiotics at all ages. In women, fluoroquinolones were used more only in patients older than 64 years. Ciprofloxacin was the most prescribed quinolone, followed by levofloxacin and norfloxacin. In a recent systematic review [31], norfloxacin was proposed as the most suitable antibiotic within this group for the treatment of pyelonephritis, with clinical cure rates comparable to those of ciprofloxacin and levofloxacin, although the activity of norfloxacin against anaerobes, streptococci and enterobacteria is lower. In our study, however, the frequency of norfloxacin use was low, at only 2%. Clinical practice guidelines [26,28,32,33] and experts [34,35] recommend the use of fluoroquinolones empirically for community-based pyelonephritis provided that the local resistance rate of does not exceed 10%. They also recommend avoiding fluoroquinolones as empirical treatment if the patient had a recent urine culture with a resistant microorganism or if fluoroquinolones were used in the past year, as this increases the risk of resistance [28].

In Spain, resistance rates during the study period were well above the recommended threshold [17], which would advise against the use of fluoroquinolones. However, some experts [13] propose that this threshold should be adapted to the reality of the clinical characteristics of patients since there are discrepancies between the resistance data and the use of certain antibiotics and their clinical response. Another controversial aspect of the use of fluoroquinolones is the presence of significant side effects and an increase of up to six times in the number of secondary infections caused by *Clostridioides difficile* [36].

The rate of amoxicillin-clavulanate prescription ~~continues~~ to be high, especially in women under 65 years of age. However, during the study period, its use was reduced by half in both sexes. It is important to highlight that prescription indications vary across countries and over time. At the time of the study, this antibiotic was not recommended as empirical treatment in Spanish clinical guidelines [30,38] or in other countries [37] due to high resistance rates. However, it was indicated for infections caused by enterococci and in pregnant women [13,37], which may explain its higher use among younger women in our study. Recent reviews and guidelines from the United States [28,34] recommend amoxicillin-clavulanate as an alternative to fluoroquinolones. In Spain [27], it is considered an alternative to third-generation cephalosporins, provided susceptibility is confirmed.

In Spain, TMP-SMX is not recommended for empirical treatment and should only be used in cases with confirmed susceptibility, for example, as an alternative to carbapenems in infections caused by extended-spectrum beta-lactamase (ESBL)-producing Enterobacteriaceae [27].

Fosfomycin was the fourth most prescribed antimicrobial in both men and women, with a higher prescription rate in women over 65 years of age. In addition, the rate of fosfomycin prescription increased throughout the study period. According to clinical practice guidelines [26,27], fosfomycin is recommended for the treatment of uncomplicated acute cystitis and should not be used for upper or severe UTIs such as pyelonephritis. One possible explanation for the high use of fosfomycin observed in our study is that it is used to treat UTIs initially assumed to be cystitis, which later progressed to fever or other signs of pyelonephritis, leading to a change in diagnosis and coding. This could also explain the higher use in older patients, where fever may be less evident than in younger adults. Another plausible explanation is the use of fosfomycin based on previous antibiograms showing multidrug-resistant organisms sensitive to this antibiotic [39] allowing for a therapeutic trial before considering hospital admission for intravenous treatment. However, some recent publications [39,40] have reported the efficacy of fosfomycin in complicated UTIs, including pyelonephritis, in selected situations.

Finally, the use of TMP-SMX was very rare, accounting for only 0.8% of prescriptions TMP-SMX was used twice as often in men as in women, and in both men and women, its use increased with age. This finding is relevant because the likelihood of resistance to this antibiotic increases with age. The low usage rate may be explained by the high resistance to TMP-SMX in Spain [17,41] exceeding 20%, a situation also reported in other countries [42–44]. Current American clinical guidelines [28] recommend TMP-SMX as an initial empirical treatment option, along with fluoroquinolones, for patients with complicated UTIs without sepsis. However, in Spain and other European countries, TMP-SMX is not recommended for empirical treatment and should only be used in cases with confirmed susceptibility, for example, as an alternative to carbapenems in infections caused by extended-spectrum beta-lactamase (ESBL)-producing Enterobacteriaceae [26,27].

## Strengths and limitations

The main strength of this study is that the data were from a population database, BIFAP, that includes real-life data over ten years. This database provides data on the incidence of pyelonephritis that have not been entered and are very rarely published, not only at the national level but also internationally. In addition, the prescription of antimicrobials obtained from the clinical records of primary care physicians in Spain represents the vast majority of the prescriptions in the National Health System.

Among the limitations of the study, there may be a problem of registration and erroneous coding, as in any database. However, the inclusion of pyelonephritis as a "case" only if it was associated with the request for an antibiogram and/or the prescription of antibiotics minimizes this problem and makes the results reliable. In addition, a previous validation of pyelonephritis diagnoses was conducted by means of a random sample analysis, which verified that there was concordance between the diagnosis and symptoms (fever, general malaise, flank pain associated with dysuria, and frequency or urgency).

It should be noted that the data are limited to the prescription of antibiotics. We cannot know if the antibiotics were dispensed and/or used. A prescription suitability analysis cannot be performed because essential data such as the dose,

duration of antimicrobial regimens, and reports on local resistance rates are missing. We also do not have data on allergies to antimicrobials or their potential impact on the use of broad-spectrum antibiotics. Comorbidity data that may influence the incidence of pyelonephritis and the choice of the prescribed antibiotic were not considered. The study did not include prescriptions by private centers or those by hospital or out-of-hospital emergency services. A lower overall prevalence of pyelonephritis is possible because only diagnoses of pyelonephritis associated with the use of antibiotics and/or requests for urine culture were considered for this study. Similarly, the antibiotics analyzed represent prescriptions for pyelonephritis treated on an outpatient basis in primary care since antibiotics prescribed for patients who required admission or were treated in other health-care settings were not considered in this study. We believe that our data on patients with community pyelonephritis treated on an outpatient basis are comparable with those of other studies that differentiate between inpatient and outpatient cases of pyelonephritis [7,8,10]

For data protection reasons, geographical data that allow geolocation are not provided, nor are the characteristics of the prescribing doctors, such as age, sex or geographic location. These data may influence the characteristics of prescriptions.

Although pyelonephritis is a relatively frequent pathology, gaps in knowledge related to it remain. It would be of great interest to develop research that provides better evidence of the incidence of community pyelonephritis, recurrences, factors that affect hospitalization and prognosis, the importance of comorbidities, the influence of drugs such as immunosuppressants and conditions such as pregnancy. In addition, studies on the adequacy of treatment in terms of the type of antibiotic chosen, dose and duration are needed. It is essential to address bacterial resistance and its association with prescribed antibiotics through periodic local studies that allow the indications for antibiotics to be updated in clinical practice guidelines.

It cannot be ruled out that another concurrent infectious process may have occurred on the same day—e.g., a respiratory infection—and that the antibiotic was used to treat both conditions. However, it would be unusual not to prioritize treatment with an antibiotic that covers pyelonephritis, given that it is a potentially serious condition. Therefore, even if another infection was present, we assumed that the antibiotic was (also) used to treat pyelonephritis.

## Conclusions

The prevalence of acute pyelonephritis at the community level in Spain is somewhat lower than that reported in other countries, and the prevalence decreased from 2009 to 2018. There were more cases of pyelonephritis in women, predominantly younger women. In men, pyelonephritis was more common among older individuals.

The most frequently prescribed antibiotics were 2nd- and 3rd-generation cephalosporins (cefuroxime and cefixime), quinolones (ciprofloxacin and levofloxacin) and penicillin derivatives (amoxicillin-clavulanic), with different profiles according to sex and age. Men received more quinolones, and women received more cephalosporins. With respect to age, cephalosporins were prescribed more often for those aged 18–64 years, and quinolones were prescribed more often for those > 65 years of age. During the study period, the rate of prescription of quinolones decreased in both sexes and age groups, whereas the rate of prescription of cephalosporins and penicillin combinations increased.

Although the trend of the use of antibiotics is close to the current recommendations such as prioritizing treatment with cephalosporins and exercising caution with fluoroquinolones, ~~these~~ results such as the high use of fosfomycin reinforce the need to understand the use of antimicrobials and prescription patterns for frequent illnesses to adapt these use patterns to the rate of local antibiotic resistance and to determine the most pertinent recommendations for each patient in context.

## Supporting information

**S1 File. RECORD checklist.**
(DOCX)

**S2 File. Antibiotic groups and active principles prescribed for acute pyelonephritis.**
(DOCX)

 

## Acknowledgments

Veronica Bryant for their essential help in the preparation of variables for accessing the BIFAP. To the colleagues of the Research Unit (Elena Polentinos, Teresa Sanz, Milagros Rico, Juan Carlos Gil and Marcial Caboblanco) for their support and help throughout the process. The authors would like to acknowledge the excellent collaboration of the primary care practitioners and pediatricians, and also the support of the regional authorities participating in the BIFAP database.

## Author contributions

**Conceptualization:** Jesus Redondo-Sánchez, Ricardo Rodríguez-Barrientos, Mª del Canto de-Hoyos-Alonso, Isabel Del Cura-Gonzalez.

**Data curation:** Jesus Redondo-Sánchez, Ricardo Rodríguez-Barrientos, Mª del Canto de-Hoyos-Alonso, Nuria Echave-Heras, Lucía Martínez-Manrique.

**Formal analysis:** Ricardo Rodríguez-Barrientos, Cristina Muntañola-Valero, Nuria Echave-Heras, Lucía Martínez-Manrique, Miguel Gil-García, Isabel Del Cura-Gonzalez.

**Funding acquisition:** Jesus Redondo-Sánchez, Ricardo Rodríguez-Barrientos, Mª del Canto de-Hoyos-Alonso, Isabel Del Cura-Gonzalez.

**Investigation:** Jesus Redondo-Sánchez, Ricardo Rodríguez-Barrientos, Mª del Canto de-Hoyos-Alonso, Miguel Gil-García, Isabel Del Cura-Gonzalez.

**Methodology:** Jesus Redondo-Sánchez, Ricardo Rodríguez-Barrientos, Mª del Canto de-Hoyos-Alonso, Isabel Del Cura-Gonzalez.

**Project administration:** Ricardo Rodríguez-Barrientos, Isabel Del Cura-Gonzalez.

**Resources:** Ricardo Rodríguez-Barrientos, Isabel Del Cura-Gonzalez.

**Software:** Ricardo Rodríguez-Barrientos, Isabel Del Cura-Gonzalez.

**Supervision:** Jesus Redondo-Sánchez, Ricardo Rodríguez-Barrientos, Mª del Canto de-Hoyos-Alonso, Miguel Gil-García, Isabel Del Cura-Gonzalez.

**Validation:** Jesus Redondo-Sánchez, Ricardo Rodríguez-Barrientos, Mª del Canto de-Hoyos-Alonso, Miguel Gil-García, Isabel Del Cura-Gonzalez.

**Visualization:** Jesus Redondo-Sánchez, Ricardo Rodríguez-Barrientos, Mª del Canto de-Hoyos-Alonso, Isabel Del Cura-Gonzalez.

**Writing – original draft:** Jesus Redondo-Sánchez, Ricardo Rodríguez-Barrientos, Mª del Canto de-Hoyos-Alonso, Miguel Gil-García, Isabel Del Cura-Gonzalez.

**Writing – review & editing:** Jesus Redondo-Sánchez, Ricardo Rodríguez-Barrientos, Cristina Muntañola-Valero, Mª del Canto de-Hoyos-Alonso, Nuria Echave-Heras, Lucía Martínez-Manrique, Miguel Gil-García, Isabel Del Cura-Gonzalez.

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
