## [Decision Letter · Decision Letter 0]

19 Aug 2025

Dear Dr. Rodríguez-Barrientos,

Thank you for submitting your manuscript to PLOS ONE. After careful consideration, we feel that it has merit but does not fully meet PLOS ONE’s publication criteria as it currently stands. Therefore, we invite you to submit a revised version of the manuscript that addresses the points raised during the review process.

We look forward to receiving your revised manuscript.

Kind regards,

Vicente Sperb Antonello, MD, MSc, Phd

Academic Editor

PLOS ONE

-----

Journal Requirements:

“This study is funded by the Instituto de Salud Carlos III (ISCIII) through project

PI19/01700 and co-funded by the European Union”

Reviewers' comments:

Reviewer's Responses to Questions

**Comments to the Author**

1. Is the manuscript technically sound, and do the data support the conclusions?

Reviewer #1: Yes

Reviewer #2: Yes

Reviewer #3: Yes

2. Has the statistical analysis been performed appropriately and rigorously?

Reviewer #1: Yes

Reviewer #2: Yes

Reviewer #3: Yes

3. Have the authors made all data underlying the findings in their manuscript fully available?

Reviewer #1: No

Reviewer #2: No

Reviewer #3: Yes

4. Is the manuscript presented in an intelligible fashion and written in standard English?

Reviewer #1: Yes

Reviewer #2: Yes

Reviewer #3: Yes

Reviewer #1: Respected Authors,

The abstract is well-organized and well-written, as is the introduction. The objective is clear. The methodology is described in detail. An appropriate statistical method has been selected. The results are clearly presented. The discussion is well-written, and the conclusion is consistent with the aim and the findings of the study.

Reviewer #2: This study provides a clear picture of the trends in pyelonephritis in primary care in Spain, as well as the patterns and changes in associated antibiotic prescribing and the demographic factors influencing this practice. Additional details are needed to better the clarify the methods and results presented.

When discussing the methods, there were a few items that were a bit unclear. When discussing the antibiotic prescriptions in the fourth paragraph of the methods section, additional clarification would be helpful regarding the "active principles" and antibiotic groups used.

In the following paragraph discussing the joinpoint analyses, additional details regarding how the analyses work would better support the use of this method. Suggest including text to note that the methodology identifies junction points without user input. From the methods and results, it was also not clear how many joinpoints could be/were identified for each analysis, and if multiple were identified, how you selected which to use. The methods also specify that the APC was estimated for each linear segment, but the junctions/joinpoints were not identified in the text to define the segments, and only the AAPCs were reported in the text. Different time periods are presented for the APCs in some figures and Table 4, but it is not specified whether these correspond to the identified joinpoints.

The titles of Figures 1, 3, and 4, presenting the joinpoint results, could be made clearer to specify which results are trends versus the results or outputs of the joinpoint analysis.

In the discussion, when mentioning that the use of amoxicillin-clavulanate remained high, it would be helpful to clarify whether this is a local guideline regarding its use, as it is commonly used for indications other than enterococci. Regarding the strengths and limitations of this work, were other diagnostic codes examined to confirm that the antibiotics prescribed were only for pyelonephritis? Prescriptions written on the same day as the diagnosis would suggest that the antibiotics were written for pyelonephritis; however, without a prescription indication field or examination of all diagnoses at the time of the primary care encounter, it cannot be confirmed that these prescriptions were for pyelonephritis and not for another indication.

Reviewer #3: POPULATION-BASED EPIDEMIOLOGICAL ANALYSIS OF ACUTE

PYELONEPHRITIS AND ANTIBIOTIC PRESCRIPTION IN SPAIN (2009-2018)

Abstract:

It is important for the authors to clarify in the text where the study took place, whether nationwide or in a specific location/region.

6.8% of the patients in the study received fosfomycin for pyelonephritis. This is very relevant, considering that fosfomycin is approved and indicated only for the treatment of uncomplicated acute cystitis in women and is not indicated for the treatment of pyelonephritis.

"Cephalosporins were the most commonly prescribed antibiotics in women, while quinolones were more common in men. An increasing trend in cephalosporin use and a decreasing in quinolone use were

observed.". It would be helpful for the authors to clarify which cephalosporins: first-generation? second-generation? third-generation?

The introduction is adequate, with no additional contributions.

The methods are clear. However, it is unclear whether the study group encompasses all autonomous regions of Spain or a limited portion of the country. Were any regions excluded or not included?

In the results section, "The most prescribed cephalosporins were cefuroxime (first generation)..." Cefuroxima is 2nd generation ceph (!!). "cefixime (second generation)", Cefixime is 3rd generation ceph (!!). Please adjust it.

In the discussion section: "One possible explanation for the use of this antimicrobial for pyelonephritis is that it is used to treat UTIs initially assumed to be cystitis. However, some recent publications (36) have

reported the efficacy of fosfomycin in complicated UTIs, including pyelonephritis..." The authors should make it clear that fosfomycin should not be prescribed to patients with pyelonephritis, based on international guidelines. Also, make state that it is not adequate. Similarly, the authors do not comment on the use of sulfamethoxazole/trimethoprim for pyelonephritis.

In this regard, I feel there is a lack of more information on the guidelines in Spain for the treatment of pyelonephritis. It would be important for the authors to describe this information clearly and objectively.

Conclusion: "Although the trend of the use of antibiotics is close to the current recommendations, these results reinforce the need to understand the use of antimicrobials and

prescription patterns for frequent illnesses to adapt these use patterns to the rate of local antibiotic resistance and to determine the most pertinent recommendations for

each patient in context." Please clarify national recommendations for pyelonephritis treatment.

**Do you want your identity to be public for this peer review?** For information about this choice, including consent withdrawal, please see our Privacy Policy

Reviewer #1: No

Reviewer #2: No

Reviewer #3: **Yes: ** Vicente Sperb Antonello

---

## [Author Response · Author response to Decision Letter 1]

9 Oct 2025

Dear Academic Editor, Vicente Sperb Antonello

Thanks for your review commentaries in order to improve the quality of the manuscript. A deep and substantial modification has been carried out according to your review suggestions. Please do not hesitate to contact us should any additional clarification be required.

Please find our responses to each reviewer comment below.

Sincerely.

Review Comments to the Author

Reviewer #1:

Respected Authors,

The abstract is well-organized and well-written, as is the introduction. The objective is clear. The methodology is described in detail. An appropriate statistical method has been selected. The results are clearly presented. The discussion is well-written, and the conclusion is consistent with the aim and the findings of the study.

Thank you very much for your comments

Reviewer #2:

This study provides a clear picture of the trends in pyelonephritis in primary care in Spain, as well as the patterns and changes in associated antibiotic prescribing and the demographic factors influencing this practice. Additional details are needed to better the clarify the methods and results presented.

When discussing the methods, there were a few items that were a bit unclear. When discussing the antibiotic prescriptions in the fourth paragraph of the methods section, additional clarification would be helpful regarding the "active principles" and antibiotic groups used.

Thank you for your valuable comment. Following your recommendation, we have revised the Methods section to clarify the concept of "active principles" and antibiotic groups. Additionally, we have included a new Supplementary File (Supplement 2) in the Results section, which provides detailed information on the antibiotic groups and active principles analyzed, as well as the most frequently prescribed ones, to enhance the clarity and usefulness of the data for readers.

Changes made:

Methods section (page 6):

Regarding the prescription of antibiotics, first, the percentage of the most prescribed antibiotics (active principles and groups of antibiotics) (all antibiotic groups and the most commonly prescribed active principles) (Supplement 2) among the total number of antibiotics prescribed during the study period was determined by age group and sex.

Results section (page 10):

Supplement 2 provides a more detailed breakdown of the usage percentages of each active principle analyzed in relation to the total number of antibiotics prescribed.

In the following paragraph discussing the joinpoint analyses, additional details regarding how the analyses work would better support the use of this method. Suggest including text to note that the methodology identifies junction points without user input. From the methods and results, it was also not clear how many joinpoints could be/were identified for each analysis, and if multiple were identified, how you selected which to use. The methods also specify that the APC was estimated for each linear segment, but the junctions/joinpoints were not identified in the text to define the segments, and only the AAPCs were reported in the text. Different time periods are presented for the APCs in some figures and Table 4, but it is not specified whether these correspond to the identified joinpoints.

Thank you for your comments regarding the analysis, which allowed us to extend the explanation of the Joinpoint method in the Methods section.

Trend analysis was conducted using Joinpoint regression. Joinpoint regression models are used to identify points in time where statistically significant changes in trend occur. The Joinpoint software identifies these change points by fitting segmented (piecewise) regression models to time series data, where each segment has a distinct slope, and the joinpoints represent changes in trend. The process does not require the user to predefine the number or location of joinpoints; instead, it uses model selection criteria such as the Bayesian Information Criterion (BIC) to determine the optimal number of joinpoints, evaluating the statistical significance of each change in slope.

In our analysis, we used the default settings, including the option to restrict joinpoints to occur at observed time points rather than between them. This allows for the identification of specific years in which trend changes occurred. All other parameters were also left at their default values, as recommended in the Joinpoint Regression Program user manual and based on the characteristics of our data.

Regarding the number of joinpoints, the software provides a graph with the segments (between years, in our study) and a list of them for each analysis, based on the user-defined settings and the statistical adjustments applied. This can be seen in the figures we have included, which were generated using the Joinpoint software.

Joinpoint programme computed graph for the lines and trend changes (by year)

Results calculated by the Joinpoint program in reference to the previous graph, which contains the corresponding segments between trend changes:

The program identified the year 2014 as a joinpoint, and it provides information about this change point and the two time segments (one from 2009-2014, other from 2014-2018) associated with it:

This is calculated by the software using the selected model selection method, which in our case was the weighted Bayesian Information Criterion (BIC) — the default setting. This method uses a combination of two metrics related to the Bayesian Information Criterion, where the weighted penalty term is based on the characteristics of the data. We chose this method because it is considered the most flexible for selecting the optimal model and is recommended by some authors (Irimata et al., 2022)*.

* Irimata, K. E., Bastian, B. A., Clarke, T. C., Curtin, S. C., & Rui, P. (2022). Guidance for selecting model options in the national cancer institute joinpoint regression software.

As shown in Table 4 and Figure 3, we present both the Annual Percent Changes (APCs) and the Average Annual Percent Changes (AAPCs) to provide a comprehensive view of the results. However, in the main text we decided to reference only one APC value: “In people older than 64 years, there was an increase in prescriptions that was significant only from 2011 to 2018 [APC 14.7 (95% CI 2.9; 25.0)]” and to focus on reporting AAPC values to facilitate interpretation for the reader. In some cases, only one APC value is available (i.e., no joinpoints were identified), and in those instances, the APC and AAPC are equivalent, so we report only one of them.

Following your suggestion, we have now added another APC value in the text that we consider relevant. The revised sentence reads:

“This decrease was significant in the group aged 18–64 years [AAPC -0.9 (95% CI -1.3; -0.4)] and was more pronounced between 2016 and 2018 [APC -2.5 (95% CI -4.6; -0.4)].”

Accordingly, the revised text for the Joinpoint methodology section, incorporating your suggestions, is as follows:

We conducted a temporal trend analysis of the incidence of acute pyelonephritis and antibiotic use during the study period using Joinpoint regression. Joinpoint regression models are widely used to identify points in time where statistically significant changes in trend occur. This approach is particularly useful when the change points (i.e., inflection points or joinpoints) are not known in advance and must be estimated from the data.

The Joinpoint software identifies change points without user input by fitting segmented (piecewise) regression models to time series data, where each segment is defined by a distinct slope, and the joinpoints represent statistically significant changes in trend. The process does not require the user to predefine the number or location of joinpoints. Instead, it uses model selection procedures such as the Bayesian Information Criterion (BIC) to determine the optimal number of joinpoints, evaluating the statistical significance of each change in slope.

In our analysis, we used the default settings, including the option to restrict joinpoints to occur at observed time points rather than between them. This allowed us to identify the specific years in which trend changes occurred. The software provides both graphical representations and detailed listings of the segments (years, in our study) for each analysis, based on the user-defined parameters and statistical adjustments.

For the trend analysis of pyelonephritis, incidence rates per 100,000 inhabitants were adjusted to the BIFAP population of the year 2014. The trend analysis of antibiotic prescriptions was conducted per 100,000 cases of pyelonephritis. We estimated the Annual Percent Change (APC) for each linear segment and calculated the Average Annual Percent Change (AAPC) as a weighted average of the APCs in the model, along with the corresponding 95% confidence intervals (95% CI).

The titles of Figures 1, 3, and 4, presenting the joinpoint results, could be made clearer to specify which results are trends versus the results or outputs of the joinpoint analysis.

Thank you for your comment, the word references multiple concepts. The joinpoint definition may cause misunderstanding. On one side, Joinpoint program is statistical software for the analysis of trends using joinpoint models, where several different lines are connected by named joinpoints (points where trend change) (Surveillance Research Program , National Cancer Institute (2025). Joinpoint Regression Software, Version 5.4.0 - April 2025)

The graphs generated represents the incidence trends of APN, calculated with joinpoint models, where de trend changes are identified (joinpoint, like APPC and APC).

Following your recommendation, we have revised the titles of Figures 1, 3, and 4 to clarify which results represent “trends” and which correspond to the outputs of the Joinpoint analysis, incorporating both terms where appropriate to improve interpretation.

The updated figure titles are as follows:

Figure 1. . Incidence trends of acute pyelonephritis and jointpoint by age group and sex in the population >18 years of age in Spain, 2009–2018

Figure 3. Incidence trends of most frequently prescribed antibiotic groups for acute pyelonephritis and jointpoint by sex in the population > 18 years of age in Spain, 2009–2018

Figure 4. Incidence trends of most-prescribed antibiotic groups for acute pyelonephritis and jointpoint by age group in the population > 18 years of age in Spain, 2009–2018

In the discussion, when mentioning that the use of amoxicillin-clavulanate remained high, it would be helpful to clarify whether this is a local guideline regarding its use, as it is commonly used for indications other than enterococci.

Thank you for your observation. We have clarified this point in the revised Discussion section. It is important to note that prescription guidelines vary over time and between countries. During the study period, amoxicillin-clavulanate was not recommended in the most widely used Spanish clinical guidelines as empirical treatment for pyelonephritis, which may help explain the trends observed in our study. We have also added a comment regarding its role in current guidelines.

The revised text in the Discussion now reads:

Remarkably, The rate of amoxicillin-clavulanate prescription continues to be high, especially in women under 65 years of age, although its indications are for infections caused by enterococci and it should not be used empirically (34,35). Notably, However, during the study period, its use was reduced by half in both sexes. It is important to highlight that prescription indications vary across countries and over time. At the time of the study, this antibiotic was not recommended as empirical treatment in Spanish clinical guidelines (30,38) or in other countries (37) due to high resistance rates. However, it was indicated for infections caused by enterococci and in pregnant women (13,37), which may explain its higher use among younger women in our study. In more recent reviews and guidelines from the United States (28, 34), amoxicillin-clavulanate is recommended as an alternative to fluoroquinolones. In Spain (27), it is considered an alternative to third-generation cephalosporins, provided susceptibility is confirmed. These changes underscore the importance of considering guideline updates when interpreting prescribing patterns.

Regarding the strengths and limitations of this work, were other diagnostic codes examined to confirm that the antibiotics prescribed were only for pyelonephritis? Prescriptions written on the same day as the diagnosis would suggest that the antibiotics were written for pyelonephritis; however, without a prescription indication field or examination of all diagnoses at the time of the primary care encounter, it cannot be confirmed that these prescriptions were for pyelonephritis and not for another indication.

Thank you for your comment. In our study, the antibiotics included were specifically those associated in the electronic health record with the pyelonephritis episode, and prescribed on day 0—that is, the same day the diagnosis was recorded.

We acknowledge that other concurrent infections (e.g., respiratory tract infections) may have been present on the same day, and that the prescribed antibiotic could have been intended to treat more than one condition. However, given the potentially serious nature of pyelonephritis, it would be clinically unusual not to prioritize an antibiotic regimen that provides appropriate coverage for this diagnosis. Therefore, even in the presence of another infection, we assumed that the antibiotic was (also) intended for the treatment of pyelonephritis.

We have added this point to the study’s limitations: pag 18

“It cannot be ruled out that another concurrent infectious process may have occurred on the same day—e.g., a respiratory infection—and that the antibiotic was used to treat both conditions. However, it would be unusual not to prioritize treatment with an antibiotic that covers pyelonephritis, given that it is a potentially serious condition. Therefore, even if another infection was present, we assumed that the antibiotic was (also) used to treat pyelonephritis.”

Reviewer #3:

Abstract:

It is important for the authors to clarify in the text where the study took place, whether nationwide or in a specific location/region.

Thank you for your comment. In the Methods section, we explain that the BIFAP database includes data from 12 autonomous communities (out of the 17 in Spain), making it representative of the Spanish population as a whole. We have now added this information to the Abstract to clarify the scope of the study.

The revised sentence in the Abstract reads:

“This is a retrospective observational population-based national study using the Database for Pharmacoepidemiological Research in the Public Domain (BIFAP), which contains primary care electronic medical records and is representative of the Spanish population.”

6.8% of the patients in the study received fosfomycin for pyelonephritis. This is very relevant, considering that fosfomycin is approved and indicated only for the treatment of uncomplicated acute cystitis in women and is not indicated for the treatment of pyelonephritis

Thank you for your comment. We agree that the use of fosfomycin in 6.8% of patients diagnosed with pyelonephritis is a relevant finding, especially considering that fosfomycin is officially approved only for the treatment of uncomplicated acute cystitis in women and is not indicated for pyelonephritis (1,2). This observation also drew our attention, and we have now highlighted it in the revised Discussion section

1. Molina Gil-Bermejo J. Pielonefritis aguda-Guía PRIOAM [Internet]. 2025. https://www.guiaprioam.com/indice/pielonefritis-aguda/

2. European Association of Urology [Internet]. [citado 19 de febrero de 2024]. EAU Guidelines on Urological Infections - Uroweb. https://uroweb.org/guidelines/urological-infections

"Cephalosporins were the most commonly prescribed antibiotics in women, while quinolones were more common in men. An increasing trend in cephalo

---

## [Decision Letter · Decision Letter 1]

23 Nov 2025

POPULATION-BASED EPIDEMIOLOGICAL ANALYSIS OF ACUTE PYELONEPHRITIS AND ANTIBIOTIC PRESCRIPTION IN SPAIN (2009-2018)

PONE-D-25-19472R1

Dear Dr. Rodríguez-Barrientos,

We’re pleased to inform you that your manuscript has been judged scientifically suitable for publication and will be formally accepted for publication once it meets all outstanding technical requirements.

Kind regards,

Vicente Sperb Antonello, MD, MSc, Phd

Academic Editor

PLOS ONE

Reviewers' comments:

Reviewer's Responses to Questions

**Comments to the Author**

Reviewer #2: All comments have been addressed

Reviewer #3: All comments have been addressed

2. Is the manuscript technically sound, and do the data support the conclusions?

Reviewer #2: Yes

Reviewer #3: Yes

3. Has the statistical analysis been performed appropriately and rigorously?

Reviewer #2: Yes

Reviewer #3: Yes

4. Have the authors made all data underlying the findings in their manuscript fully available?

Reviewer #2: Yes

Reviewer #3: Yes

5. Is the manuscript presented in an intelligible fashion and written in standard English?

Reviewer #2: Yes

Reviewer #3: Yes

Reviewer #2: (No Response)

Reviewer #3: All comments have been addressed adequately and have no further questions to ask for the authors of the study.

**Do you want your identity to be public for this peer review?** For information about this choice, including consent withdrawal, please see our Privacy Policy

Reviewer #2: No

Reviewer #3: **Yes: ** Vicente Sperb Antonello

---

## [Editor Report · Acceptance letter]

PONE-D-25-19472R1

PLOS One

Dear Dr. Rodríguez-Barrientos,

I'm pleased to inform you that your manuscript has been deemed suitable for publication in PLOS One. Congratulations! Your manuscript is now being handed over to our production team.

Kind regards,

on behalf of

Dr. Vicente Sperb Antonello

Academic Editor

PLOS One